# Public Anxiety, Attitudes, and Practices towards COVID-19 Infection in the Eastern Province of Saudi Arabia: A Cross-Sectional Study

**DOI:** 10.3390/healthcare11142083

**Published:** 2023-07-21

**Authors:** Mahmoud Mohamed Berekaa, Abdulaziz Abdulrahman AlMulla, Munthir Mohammed AlMoslem, Khalid Saif AlSahli, Mohammed Tawfiq AlJassim, Abdulmalik Salman AlSaif, Salman Ali AlQuwayi

**Affiliations:** Department of Environmental Health, Collage of Public Health, Imam Abdulrahman Bin Faisal University (IAU), P.O. Box 1982, Dammam 31441, Saudi Arabia; aaalmulla@iau.edu.sa (A.A.A.); mmalmoslem@iau.edu.sa (M.M.A.); ksalsahli@iau.edu.sa (K.S.A.); mtaljassim@iau.edu.sa (M.T.A.); aalsaif@iau.edu.sa (A.S.A.); saalquwayi@iau.edu.sa (S.A.A.)

**Keywords:** COVID-19 pandemic, coronavirus, non-therapeutic interventions, vaccination, chronic diseases, anxiety, attitude

## Abstract

Coronavirus disease 2019, or COVID-19, still has a terrifying potential due to its continuous genetic variation. Although vaccines have been created, adherence to preventive measures remains a privileged choice to tackle the pandemic. This study aims to investigate the anxiety, attitudes, and practices (KAPs) towards COVID-19 infection in the Eastern Province of Saudi Arabia. In this cross-sectional study, data were collected from 400 participants via an online self-structured questionnaire. Anxiety, attitude, and practice scores were calculated by summing the points of the statements under the corresponding domain multiplied by 100 over 12, 21, or 15, respectively. Chi-square and one-way analysis of variance were used to investigate the relationships between vaccination, anxiety, attitude, practice scores, and demographic characteristics. More than half of the participants were female (58.5%; mean age of 29.5 years; the majority in the age groups of <20 years and 21–30 years). Only 21.5% of the participants were suffering from or previously had chronic diseases. Notably, 22.3% of the male participants were vaccinated (*p* = 0.000). The old age groups (41–50 years and >50 years) were more vaccinated (16.3% and 24.1%, respectively, *p* = 0.000), as well as the unemployed (36.4%; *p* = 0.000). The mean scores of anxiety, attitude, and practice were 66.8, 72.3, and 85.2, respectively. Females had an anxiety score of 68.5% (*p* = 0.008) and a higher attitude score of 68.5% (*p* = 0.008). Infected male participants had a lower practice score of 80% (*p* = 0.038), while females recorded higher practice scores (85.7 ± 11.6). The results highlight the importance of reliable communication from health representatives and legislators in educating the public and promoting their knowledge about non-therapeutic interventions. Efficient intervention approaches are required to fill the gap during the implementation of non-therapeutic measures. Also, it is recommended that awareness programs, during COVID-19 or any other similar pandemics, should be tailored to target Eastern Province inhabitants, especially males.

## 1. Introduction

Severe acute respiratory syndrome coronavirus 2 (SARS-CoV-2), previously known as coronavirus disease 2019 (COVID-19), is a disease caused by a member of RNA viruses of a probable bat origin, discovered early in Wuhan, Hubei Province, China [1,2]. COVID-19 has shown a vast range of clinical manifestations, ranging from asymptomatic to life-threatening symptoms. The symptoms of COVID-19 include fever, dry cough, fatigue, myalgia, and shortness of breath [3,4]. Recently, several therapeutic strategies have been implemented for the development of an efficient therapy against COVID-19 infection; however, most of these therapies lack specificity and effectiveness. Moreover, continuous variation in the genetic material during viral replication has led to the emergence of more potent variants that frightened the world [5,6]. Therefore, public adherence to preventive measures remains the best choice for the control and prevention of viral spreading [7]. According to Johnson and Hariharan [8], providing health information to the public during outbreaks is a key component of an efficient outbreak control strategy. Public knowledge, attitudes, and practices (KAPs) can improve communication and mitigation strategies during pandemics as well as support future pandemic preparedness planning. Therefore, it is important to tailor awareness programs among populations during a pandemic due to differences in the nature, causes, control measures, and management of different pandemics [8].

The realisation of any approach to fighting COVID-19 heavily depends on public adherence to the guidelines published by the appropriate agency [9]. Hence, it is essential to appreciate the KAPs of the community to explore and guarantee the realisation of administration policy to battle this pandemic. KAP assessments are commonly used to recognise knowledge gaps and behaviour patterns among various sociodemographic subcategories to execute efficient public health mediations and interventions [10].

On the global level, studies on non-therapeutic interventions to combat the spread of COVID-19 among the public have been recorded in many countries, including India, Malaysia, the Philippines, South Korea, China, Bangladesh, Sudan, and Indonesia [11,12,13,14,15,16,17,18]. The main target of these studies is to reconcile the relationship between knowledge and the major precautionary activities, including wearing face masks, practicing hand hygiene, social distancing, and avoiding mass gatherings.

On a national level, significant studies have been conducted to address the role of KAPs among Saudi populations in combating COVID-19 infections during the pandemic [19,20,21,22,23,24]. Moreover, the realisation of these non-therapeutic interventions is determined by the public’s KAPs to tackle the pandemic [25,26,27]. On the other hand, gender differences in KAPs towards COVID-19 infection have been investigated. Alahdal [20] revealed that males showed a slight increase in awareness level compared with females, while females showed better practices towards COVID-19. Recently, Anaam [28] recognised that females recorded a higher level of knowledge than males, especially in avoiding crowded places and following precautionary measures. The incidence of general anxiety disorder (GAD) towards COVID-19 infection was significantly increased among participants, including females and individuals with chronic diseases [29,30].

Assessing public knowledge is highly significant in identifying gaps, improving control and risk management, and strengthening practices in the implementation of prevention measures during a pandemic. Therefore, this study aims to examine the Saudi community’s attitudes and practices towards COVID-19 infection in the city of Dammam, Eastern Province, Saudi Arabia. Emphasis was given to possible anxiety due to the pandemic, reactions towards possible infected citizens, and daily practices during the pandemic.

## 2. Materials and Methods

### 2.1. Research Design and Data Collection

This study was conducted in Dammam, Easter Province, Saudi Arabia, between February and March 2021. The results in Figure 1 show the distribution of cases during the study period. Although the number of positive cases increased, it was clear that the number of recovered cases was even higher, with very few death cases.

This cross-sectional study assessed the level of anxiety, attitude, and practice towards COVID-19 infection among adult residents in Dammam, Eastern Province, Saudi Arabia. The sample size was calculated using Cochran’s formula [31]. Data were collected from participants via an online questionnaire survey (using Google Forms), which was distributed through social media (WhatsApp, Telegram, and Twitter) using the convenience sampling technique. All adult (>18 years old) Dammam residents who can use media platforms and understand the content of the questionnaire were eligible to participate in the questionnaire survey for this study. Individuals unwilling to participate had the right to leave the survey at any time. Participants unable to understand the content of the questionnaires were excluded.

The questionnaire comprised 22 questions divided into different parts. The first part included 5 questions related to sociodemographic data (age, gender, city of residence, number of family members, and health status), and the second part included 17 COVID-19-related questions (personal and family history of COVID-19 infection, awareness degree of the main symptoms of COVID-19, attitude towards COVID-19 infection, and commitment to protective measures). The questionnaire was written in Arabic, and three public health experts determined and confirmed the face and content validity of the questionnaire. Before administering the survey, Cronbach’s alpha test was performed to check the reliability of the questionnaire, which was found to be moderate (0.61) [32]. Although this value has a potential degree of uncertainty, it is still mostly acceptable, as mentioned by Daud et al. [33]. In addition, ethical approval was obtained from the Institutional Review Board (IRB) of Imam Abdulrahman Bin Faisal University (IRB-2021-03-020).

### 2.2. Data Analysis

Initially, the data were collected and recorded in an Excel sheet and then exported to the Statistical Package for the Social Sciences program, version 25, for analysis. The quantitative variables were presented in the form of mean and standard deviation, whereas the categorical variables were presented in numbers and percentages. The anxiety score was calculated by summing the points of the statements under this domain multiplied by 100 over 12. The attitude score was calculated by summing the points of the statements under this domain multiplied by 100 over 21. The practice score was calculated by summing the points of the statements under this domain multiplied by 100 over 15. The chi-square test was used to evaluate the relationship between the vaccination variable (yes, no) and the demographic characteristics. One-way analysis of variance and *t*-test were used to determine the differences in the anxiety, attitude, and practice scores with regard to demographic characteristics. *p*-value < 0.05 (95% confidence interval) was considered statistically significant.

## 3. Results and Discussion

### 3.1. Demographic Characteristics of Participants

The study included 400 citizens from the Eastern Province of Saudi Arabia who responded to the online questionnaire. Table 1 shows the demographic characteristics of the participants. More than half of the participants were females (58.5%), and 41.5% were males. The mean age was 29.5 years, with a majority in the age groups of <20 years and 21–30 years (35% and 28%, respectively). Regarding occupation, the majority were either employed 51.8% or students 45.5%, and only 2.8% were unemployed, reflecting the importance of KAPs among those groups to combat COVID-19 infection and disease. In addition, 21.5% of the participants were suffering from or previously had chronic diseases (diabetes, heart, blood pressure, and asthma). Generally, the increased expression of ACE2 due to the treatment of diabetes and hypertension increases susceptibility to COVID-19 and therefore increases the risk of acute and lethal COVID-19 infection [30]. Furthermore, 5% of the participants were infected, and 10.5% of them received vaccination. Interestingly, these findings align with the results obtained by Al-Hanawi et al. [19]. The low number of COVID-19-infected respondent citizens from Eastern Province in Saudi Arabia during the period of study (1.31% of all infected citizens in the kingdom) and throughout the kingdom (Figure 1) might be due to extraordinary relaxation during the public lockdown and steady move towards stringent intelligent lockdown in Saudi Arabia [34,35,36,37]. In addition, it may possibly be due to a decline in COVID-19 death cases in Saudi Arabia compared with other countries.

### 3.2. Response towards Vaccination

Generally, vaccine hesitancy and the psychological status of participants towards vaccination are significant factors that could enhance or hinder community response during planning for effective health communication to boost mass immunisation [38].

Table 2 shows the relationship between vaccination status and demographic characteristics. Males were vaccinated more than females (22.3% and 2.1%, respectively), and the relationship was statistically significant (chi = 41.9, *p*-value = 0.000). These results reflect the significant acceptance and trust towards vaccination among males. In contrast, Nour [39] reported that this significance was higher among the female population in Makkah (OR = 1.62; CI: 1.1–2.39 for acceptance and OR = 4.15; CI: 2.86–6.04 for trust).

The older age groups of 41–50 years and >50 years were more vaccinated than the young age groups (16.3% and 24.1%, respectively), whereas the percentages of the vaccinated persons in the age groups of <20 years, 21–30 years, and 31–40 years were 0.7%, 14.3%, and 14.9%, respectively. Thus, there was a statistically significant relationship between age and being vaccinated (chi = 24.6, *p*-value = 0.000). In alignment with this finding, Mohaithef and Padhi [40] reported that the willingness to accept the COVID-19 vaccine is relatively higher among older age groups.

Unemployed participants were found to be more vaccinated (36.4%) than students (3.8%) and employees (15%), and the relationship was statistically significant (chi = 12.5, *p*-value = 0.000). In contrast, Mohaithef and Padhi [40] revealed that citizens with higher degree education levels are more committed to accepting vaccines (68.8%). Moreover, although unemployed participants were vaccinated in the current study, Mohaithef and Padhi [40] reported that employed participants are more motivated to accept the vaccine.

Interestingly, a statistically significant relationship was observed between being vaccinated and suffering from chronic diseases, as 16.3% of those with chronic diseases were vaccinated, while only 8.9% of those with no chronic diseases were vaccinated (chi = 3.89, *p*-value = 0.048).

McIntyre [41] indicated that COVID-19 vaccination approaches should focus on people with severe diseases together with global equity as a priority to minimise COVID-19 infection from the five known variants [41]. Interestingly, the willingness of adults and elderly people with chronic diseases, such as cancer, kidney disease, lung disease, diabetes, dementia, obesity, and heart conditions, to receive the vaccine in China and Saudi Arabia was recognisable [42,43]. On the other hand, it is worth noting that in the current study, we did not find a statistically significant relationship between being vaccinated and being infected (chi = 0.678, *p*-value = 0.410). Finally, during the study period, only 10.5% of the participants got vaccinated, reflecting the importance of health education intervention in increasing people’s commitment to accept vaccines during pandemics.

### 3.3. Anxiety, Attitudes, and Practices towards COVID-19 Infection

In the current study, the scoring system used for the assessment of anxiety, attitudes, and practices among participants, shown in Table 3, revealed that the mean anxiety score among the participants was 66.8 SD ± 15.2, 72.3 SD ± 10.4 for the attitude score, and 85.2 SD± 11.5 for the practice score.

### 3.4. Anxiety and COVID-19 Infection

On the other hand, pandemic outbreaks, including COVID-19, raise global anxiety and fear of infection. Therefore, for the success of non-pharmaceutical interventions among the public, the assessment of social, mental, and psychological aspects and behaviours is essential. Generally, mental health examinations during the pandemic could shed light on strategies that could be provided by health authorities whenever required [13]. Differences in anxiety scores relative to demographic characteristics in this study are presented in Table 4. Among the examined variables, only gender showed differences in the anxiety score. Females had an anxiety score of 68.5 SD ± 14.8, which is more than males, at 64.4 SD ± 15.6 (t = 2.67, *p*-value = 0.008).

Similarly, Lin [13] revealed that females were significantly anxious regarding COVID-19 infection, as recorded by the STAI-S questionnaire for state and trait anxiety (OR = 1.67) and STAI-T (OR = 1.30) in China. In addition, Malesza [44] revealed that women reported higher anxiety about COVID-19. In Saudi Arabia, depression, anxiety, and stress were significantly higher among females, and they demonstrated higher anxiety levels together with depression and stress during the COVID-19 pandemic [45]. Alsaif [29] revealed that the frequency of GAD was 21.8% and was significantly increased among citizens from the Hail community in Saudi Arabia, especially among those with poor knowledge (OR = 2.01; 95% CI: 1.25–3.22), female candidates (OR = 1.92; 95% CI: 1.19–3.09), participants harbouring chronic diseases (OR = 1.71; 95% CI: 1.02–2.86), and non-Saudi participants (OR = 2.44; 95% CI: 1.02–5.85). Moreover, higher anxiety levels were recorded among female healthcare workers and female students (OR = 1.963, 95% CI = 1.160, 3.322, *p*-value = 0.012, respectively) [46,47].

Participants who were unvaccinated had more anxiety (67.3 SD ± 15.2) than those who were vaccinated (62.7 SD ± 14.9), but the difference was not statistically significant (*p*-value = 0.064). Islam [48] reported that anxiety was present among participants who intended to take a vaccine. Moreover, a bidirectional association between vaccine hesitancy and mental health was observed in the Kingdom of Saudi Arabia (KSA) [49]. On the other hand, chronically diseased candidates usually have a high level of anxiety or GAD due to lower life standards and an elevated level of psychological distress, especially due to the inability to follow their regular medical care during the COVID-19 pandemic [50,51,52]. This was not observed among Dammam inhabitants. Moreover, there was no statistically significant difference between anxiety scores regarding the other examined variables.

### 3.5. Attitudes and Behaviours towards COVID-19 Infection

Participants’ attitudes towards COVID-19 infection, such as willingness to attend events, touching surfaces, obeying the precautionary measures, and adhering to the precautionary measures after recovery were investigated and assessed. Differences in attitude scores relative to demographic characteristics are presented in Table 4. The results showed a significant positive attitude towards COVID-19 infection (72.3 SD ± 10.4). Similarly, there was an optimistic attitude and good practices among the Saudi Arabia population [19]. In a previous study, it was revealed that most Malaysian residents followed precautionary measures, such as avoiding crowds (83.4%) [11]. In contrast, the majority of the Filipino population did not obey the rules of social distancing and avoiding mass gatherings (32.4% and 40.6%, respectively) [17]. In terms of the correlation between demographic characteristics and attitudes, the only variable that showed differences in the attitude score was gender. Females had an attitude score of 68.5, which is higher than males at 64.4 (t = 2.67, *p*-value = 0.008; Table 4). Similarly, Galasso [53] revealed serious perceptions of COVID-19 among females in eight countries, concluding that communication on COVID-19 for the implementation of public health policies should be based on gender. Conversely, a study conducted by Ferrín [54] showed that women were more reluctant than men to adhere to preventive behaviours against COVID-19. Moreover, in India, attitudes assessment indicated that males had a higher score than their female counterparts [16]. However, there was no statistically significant difference between anxiety scores regarding the other examined variables (age, occupation, suffering from chronic diseases, if infected, and if vaccinated). Instead, Noushad [55] reported that adults recorded a higher level of motivation for vaccination due to increased levels of anxiety (*p*-value < 0.05). In addition, the majority of the Saudi participants showed positive attitudes and perceptions towards the use of vaccines [56].

### 3.6. Practices towards COVID-19 Infection

As recommended by the World Health Organization and the Ministry of Health in Saudi Arabia, people should take several precautionary measures to minimise the risk of infection and to reduce viral dissemination, including social distancing, wearing masks, reducing hand shaking, and washing hands. The results shown in Table 4 indicate that all of the participants showed significant practice scores, principally infected and non-infected participants (80.0 SD ± 11.8 and 85.2 SD ± 11.2, respectively). Siddiqui [24] revealed that 79% of participants in Saudi Arabia realised the importance of safe distancing. During the early pandemic, Alnasser [23] revealed that 94.3% of Saudi participants avoided crowded places, similar to a study conducted in India and China [57,58]. In addition, Singh [16] revealed that social distancing was among the three major preventive measures used by Indians to combat the COVID-19 pandemic. In contrast, in this study, considering the differences in practice scores in terms of demographic characteristics, which are presented in Table 4, the only variable that showed differences in practice scores was “if infected”. Those who had been infected had a practice score of 80, which is lower than those who had not been infected, at 85.1 (t = 2.077, *p*-value = 0.038). In Riyadh, 99% of participants agreed on handwashing and mask wearing as helpful practices [20]. Siddiqui [24] revealed that 84% of participants in Saudi Arabia realise the importance of handwashing. Alnasser [23] revealed that 86.4% of Saudi participants considered face masks an important measure to practice self-control during the pandemic, and 93.8% wear masks when they go outside. Similarly, 68.8% of Pakistani always wear a face mask when leaving their homes, while 27% do not [59]. In concordance, a similar practice has been reported in other countries such as India, Vietnam, and Ecuador [57,60,61].

Similarly, Alnasser [23] revealed that 91.85% of Saudi participants practiced hand hygiene. In concordance, most Malaysian residents practiced proper hand hygiene (87.8%) [11]. Even in developed countries, e.g., the Philippines, handwashing was recognised by 82.2% of participants as a precautionary measure to combat the virus [17]. Although gender was not statistically different, females recorded higher practice scores (85.7 SD ± 11.6). Moreover, Al-Hanawi [19] revealed that women generally have better practices than men. Alahdal [20] revealed that although males showed a slight increase in awareness level compared with females (60% vs. 57%), practice scores revealed that females showed better practices towards COVID-19 (82% vs. 80%). Recently, Anaam [28] examined the level of knowledge, attitudes, and practices among Saudi Arabian males and females towards COVID-19. Interestingly, it was recognised that females recorded higher knowledge than males (82.4% vs. 78.5%, *p*-value = 0.005), especially in avoiding crowded places (57.7% vs. 46.2%, *p*-value = 0.04), and approximately 84.4% still follow precautionary measures (89.9% vs. 79.5%; *p*-value = 0.01). Moreover, unlike males, female participants recorded positive, significant scores regarding their knowledge of and practices towards COVID-19. In the Indian population, the mean practice score was 10.3, with significant differences among females (score 10.6) [16]. Thus, it is recommended that awareness programs should be tailored to specifically target males.

The current study highlights the crucial significance of non-pharmaceutical measures to combat COVID-19 and similar diseases during the early stages of pandemics. The precautionary measures being implemented by the Saudi government and healthcare agencies explain the remarkably decreased number of infected participants in comparison with many other countries. In this study, we assessed the anxiety, attitudes, and practices towards COVID-19 among infected individuals, in comparison with non-infected and other high-risk groups. Also, a major key feature that distinguishes the current study from others is the assessment of anxiety among COVID-19-infected and non-infected participants in relation to demographic characteristics. Interestingly, based on the results of the current study, females showed higher anxiety scores than males. Also, public acceptance of vaccines was closely assessed, and the results of the study showed that older male groups were more vaccinated than the same age group of females. In addition, the study unravels the relationship between being vaccinated and suffering from chronic diseases. Moreover, the study highlights the importance of targeted health education intervention to increase public commitment to accept vaccines for future COVID-19 outbreaks and other similar pandemics. Generally, there were optimistic attitudes and good practices among the Easter Province population in Saudi Arabia.

On the other hand, there are some limitations that should be considered in the interpretation of the results. Firstly, features related to psychological fear or anxiety among infected participants should also be considered in the early stages of the pandemic. Secondly, the response and trust of participants towards vaccines might change during the context of pandemics due to massive governmental vaccinations and media interventions. Thirdly, online questionnaires are mostly available for young people; therefore, there is a potential for participation bias. Finally, the results of the KAP assessment of the Dammam community in Easter Province cannot be generalised to the national level.

## 4. Conclusions

Due to continuous variation in the COVID-19 genome, reliance on non-therapeutic intervention and application of preventive measures remains an efficient strategy to mitigate the risks associated with COVID-19 infection. The success of these non-therapeutic interventions is dependent on public anxiety, attitudes, and practices to combat the disease. Inefficient practices and attitudes, with increased anxiety, might increase the risk of viral transmission among citizens. The current study highlights the limited number of infections due to efficient governmental policies during the pandemic. Participants from the Eastern Province region had acceptable levels of attitudes and practices regarding COVID-19 infection. Males were vaccinated more than females, which indicates a significant level of acceptance and trust among males towards vaccination, especially among older age groups (41–50 years and >50 years) and employed people (36.4%). A statistically significant relationship was also shown between being vaccinated and suffering from chronic diseases. Indeed, targeted health education intervention is recommended to increase public commitment to accept vaccines for future COVID-19 outbreaks and other similar pandemics. In the current study, females showed higher anxiety scores than males. There were optimistic attitudes and good practices among the Saudi population. Only infected participants revealed differences in practice scores. This study provides a foundation for the development of awareness programs to enhance preventive measures and non-therapeutic interventions among the public. Furthermore, our understanding of the existing public awareness and practices can be further improved. Thus, enriching public knowledge to recognise characteristics that influence the community to embrace beneficial practices and adhere to preventive behaviours is of great importance.

## Figures and Tables

**Figure 1 healthcare-11-02083-f001:**
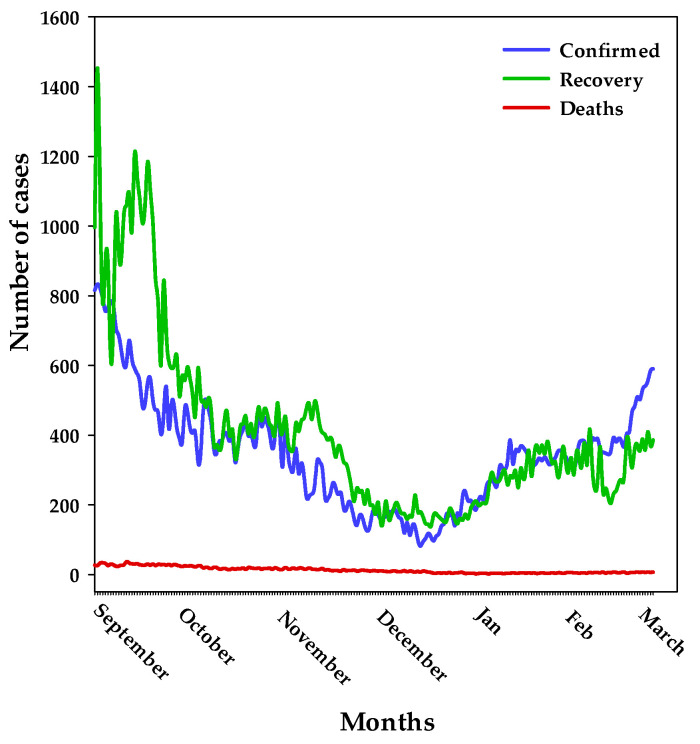
Distribution of COVID-19 cases during the study period.

**Table 1 healthcare-11-02083-t001:** Demographic characteristics of participants.

Variable	Number (N)	Frequency (%)
**Gender**		
Male	166	41.5
Female	234	58.5
**Age** Mean ± SD (min–max) 29.5 ± 11.8 (16–70)		
<20 years	140	35.0
21–30 years	112	28.0
31–40 years	74	18.5
41–50 years	43	10.8
>50 years	29	7.3
NA	2	0.5
**Occupation**		
Student	182	45.5
Not employed	11	2.8
Employed	207	51.8
**Do you suffer or have ever had a chronic disease (diabetes, heart, blood pressure, asthma)?**		
No	314	78.5
Yes	86	21.5
**Are you infected?**		
No	380	95.0
Yes	20	5.0
**Did you get the vaccine?**		
No	358	89.5
Yes	42	10.5

**Table 2 healthcare-11-02083-t002:** Relationship between demographic characteristics and being vaccinated.

Variable	Vaccinated			Chi	*p*-Value
	No (%)	Yes (%)	Total (%)		
**Gender**					
Male	129 (77.7)	37 (22.3)	166 (100)	41.967	0.000 *
Female	229 (97.9)	5 (2.1)	234 (100)		
**Age**					
<20 years	139 (99.3)	1 (0.7)	140 (100)	24.6	0.000 *
21–30 years	96 (85.7)	16 (14.3)	112 (100)		
31–40 years	63 (85.1)	11 (14.9)	74 (100)		
41–50 years	36 (83.7)	7 (16.3)	43 (100)		
>50 years	22 (75.9)	7 (24.1)	29 (100)		
**Occupation**					
Student	175 (96.2)	7 (3.8)	182 (100)	12.5	0.000 *^#^
Not employed	7 (63.6)	4 (36.4)	11 (100)		
Employed	176 (85)	31 (15)	207 (100)		
**Do you suffer or have ever had a chronic disease (diabetes, heart, blood pressure, asthma)?**					
No	286 (91.1)	28 (8.9)	314 (100)	3.893	0.048 *
Yes	72 (83.7)	14 (16.3)	86 (100)		
**Are you infected?**					
No	339 (89.2)	41 (10.8)	380 (100)	0.678	0.410
Yes	19 (95)	1 (5)	20 (100)		

* Statistically significant, ^#^ Fishers’ test.

**Table 3 healthcare-11-02083-t003:** Descriptive analysis of anxiety, attitude, and practice scores.

Variable	Mean ± SD	Min–Max
Anxiety score	66.8 ± 15.2	33.3–100
Attitude score	72.3 ± 10.4	42.9–95.2
Practice score	85.2 ± 11.5	10–100

**Table 4 healthcare-11-02083-t004:** Differences in anxiety, attitudes, and practices with regard to demographic characteristics.

Variable	Number	Anxiety	*p*-Value	Attitude	*p*-Value	Practice	*p*-Value
		Mean ± SD		Mean ± SD		Mean ± SD	
**Gender**							
Male	166	64.4 ± 15.6	0.008 *	70.9 ± 11	0.025 *	84.4 ± 11.4	0.273
Female	234	68.5 ± 14.8		73.3 ± 9.9		85.7 ± 11.6	
**Age**							
<20 years	140	68.1 ± 14.4	0.102	73.5 ± 9.7	0.213	86 ± 12.4	0.330
21–30 years	112	64.4 ± 15.8		70.5 ± 11.6		84.1 ± 11.1	
31–40 years	74	65 ± 17.1		71.9 ± 11.1		83.7 ± 11.7	
41–50 years	43	70 ± 13.6		73 ± 9.3		85.6 ± 10	
>50 years	29	69.8 ± 13.3		73.4 ± 9.1		88.1 ± 10	
**Occupation**							
Student	182	67.8 ± 14.9	0.460	73.3 ± 10.3	0.217	85.6 ± 12.5	0.831
Not employed	11	64.4 ± 13.5		70.6 ± 9.7		84.8 ± 9	
Employed	207	66.1 ± 15.6		71.5 ± 10.6		84.9 ± 10.7	
**Do you suffer or have ever had a chronic disease (diabetes, heart, blood pressure, asthma)?**							
Yes	86	65.2 ± 16.7	0.273	70.6 ± 11.5	0.093	85.6 ± 11.5	0.719
No	314	67.3 ± 14.8		72.7 ± 10.1		85.1 ± 11.5	
**Are you infected?**							
Yes	20	67.5 ± 15	0.836	71.4 ± 12	0.711	80 ± 11.8	0.038
No	380	66.8 ± 15.3		72.3 ± 10.4		85.5 ± 11.4	
**Did you get the vaccine?**							
Yes	42	62.7 ± 14.9	0.064	70 ± 11.4	0.128	84.3 ± 11.9	0.593
No	358	67.3 ± 15.2		72.5 ± 10.3		85.3 ± 11.5	

* Statistically significant.

## Data Availability

Not applicable.

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
