# Peer review of "Public Anxiety, Attitudes, and Practices towards COVID-19 Infection in the Eastern Province of Saudi Arabia: A Cross-Sectional Study"

_healthcare, 2023, doi:10.3390/healthcare11142083_

Round 1
Reviewer 1 Report
Thank you for inviting me to review this article. In the attached document you will find the revision of the manuscript.
Best regards.

Author Response
Reply to Review-1 Comments
Journal: Healthcare (ISSN 2227-9032)
Manuscript ID: healthcare-2474892
Type: Article
Title: Public Anxiety, Attitudes, and Practices towards COVID-19 Infection in Dammam, Saudi
Arabia: A Cross-sectional Study.
Authors: Mahmoud Mohamed Berekaa, Abdulaziz Abdulrahman AlMulla, Munthir Mohammed AlMoslem, Khalid Saif AlSahli, Mohammed Tawfiq AlJassim, Abdulmalik Salman AlSaif, Salman Ali AlQuwayi.
Point-by-point reply to reviewer-1 comments:
Reviewer-1:
First of all, thank you for inviting me to review this article. The main objective of the article was to investigate the Eastern Province resident’s anxiety, attitude and practices towards COVID-19 infection. Congratulations on the subject of the study. Beyond the physical implications of COVID, it is vitally important to analyse how it affects our mental health and behaviour, especially at a time when mental health is so compromised. The analysis of such a specific population is probably conditioned by its biopsychosocio-cultural determinants, giving us a more accurate view of its reality. After reviewing the article, I will proceed to make a series of considerations for each of the sections of the manuscript. I would like to point out that although some modifications to the manuscript are necessary, the writing and execution of the study is very good. It is one of the clearest and most correct investigations I have read. Congratulations.
Reply:
Thanks for your comment.
Title and keywords.
Reviewer1:
Nothing to contribute, the information is correct and adequate. They identify the object of study, the population, and the design in a clear, concise and attractive way.
Reply:
Thanks for your comment.
Abstract.
The information in this section is correct, nothing to contribute. Congratulations on a well-structured abstract with adequate information.
Reply:
Thanks for your comment.
Introduction.
Congratulations, you have made a clear and well-structured introduction. It puts the subject in context perfectly. I would like to point out some aspects that could be modified.
- Add some data on COVID in the study region, to justify the adoption of the measures.
Reply:
More information was added about number of infected cases in the study area during the period of the study (with reference).
The total number of cases in the study area was 6618 (out of 47235) during the period of the study, that represents 1.31% of all cases in the kingdom during that period.
- Lines 46 and 47 add changes to the wording. As reported by [8], providing health information during outbreaks to the public is a key component for efficient outbreak control strategy.
- For example, "According to Johnson et al. (2017) [8] providing health information to the public during outbreaks is a key component of an effective outbreak control strategy" or "Thus, providing health information to the public during outbreaks is a key component of an effective outbreak control strategy [8]".
Reply:
Thank you for your comment. The following modification was done “According to Johnson and Hariharan [8], ……..”
- In line 48 it refers to KAP without specifying its meaning, which it specifies in lines 53-54. Please indicate its meaning in line 48 to enable a smooth understanding of the text.
Reply:
The meaning of KAP was specified as recommended.
Materials and methods.
The wording of this section allows the research to be reproduced perfectly. Congratulations.
The reliability of your questionnaire (Cronbach's alpha = 0.61) is moderate and although public health experts endorse your questionnaire, you could provide an indicator to support or justify this reliability. I am aware that the Cronbach's alpha is a measure of internal consistency and does not address other important aspects of the quality of a questionnaire, such as content validity or construct validity. Thank you very much.
Reply:
Thanks for your comment. Additional data concerning the quality of the questionnaire was added. In fact, the validity of the questionnaire was determined by face validity and content validity, that were confirmed by three experts in public health. For reliability only Cronbach’s alpha was determined. Although alpha Cronbach value was 0.6 i.e. between 0.60 - 0.80 is considered moderate, but mostly acceptable as mentioned in Daud et al., 2018. (https://isdsnet.com/ijds-v7n3-15.pdf).
Daud, K.A.M., Khidzir, N.Z., Ismail, A.R. and Abdullah, F.A. (2018), “Validity and reliability of instrument to measure social media skills among small and medium entrepreneurs at Pengkalan Datu River”, International Journal of Development and Sustainability, Vol. 7 No. 3, pp. 1026-1037.
Results.
I congratulate you on your results, which are clearly and concisely written. This section is only to present the results obtained. It does not give room for comparison with other studies and research, please remove that part and place it in the discussion.
Reply:
First, thank you very much for your comment on the results section, especially for its clarity and conciseness.
- On the other hand, results and discussion were merged into one section.
- We assume that it is advantageous to merge results and discussion because each section in the results and discussion treated independently. Moreover, discussion of the results and findings in the same section would have greater impact than separation of them. Where, researchers can see the results, its presentation and enriched discussion in comparison with/to other researchers’ findings at the same section.
- Moreover, as mentioned in Instructions for Authors related to Healthcare journal, it is stated that there is possibility to combine discussion with the results as following:
- Discussion: Authors should discuss the results and how they can be interpreted in perspective of previous studies and of the working hypotheses. The findings and their implications should be discussed in the broadest context possible and limitations of the work highlighted. Future research directions may also be mentioned. This section may be combined with Results.
https://www.mdpi.com/journal/healthcare/instructions
- Moreover, discussion of results was stated point-by-point at the end of each result section as in the following examples:
3.1. Demography and Characteristics of Participants
Discussion started from “Generally, the increased expression of ACE2 ………….. to ……… in the direction of stringent intelligent lockdown in Saudi Arabia [31,32].”
3.2. Response towards Vaccination
Discussion started from “McIntyre [37] discussed that COVID-19 vaccination …………. to ……………… in China and Saudi Arabia was recognizable [38,39].
3.4. Anxiety and COVID-19 Infection
Discussion started from “Similarly, Lin [13] revealed that females are …………. to ……………… female students (OR = 1.963, 95% CI = 1.160, 3.322, p-value = 0.012, respectively) [42,43]. Also, “Islam [44] reported that anxiety was present among ……. to ……… COVID-19 pandemic [46–48], this was not recognized among Dammam inhabitants.
Discussion.
This section is missing. This is intended for comparison of your data with that of other research, which you did in the previous section, place it in this section.
Reply:
As previously mentioned during reply to the above question. The results and discussion were merged in one section. This format is acceptable by the Healthcare Journal instructions for authors. Moreover, detailed discussion of each result in the study was intensively discussed at the end of each sub-section. Please see the above examples 3.1., 3.2. and 3.4. as well as other sub-sections in the manuscript.
Conclusions.
Nothing to contribute, the information is correct and adequate.
Reply:
Thank you for your comment.
References.
The references used in this manuscript are appropriate and no unnecessary self-citations have been detected. Congratulations on the use of up-to-date references, most of which are less than 5 years old.
Reply:
Thank you for your comment.

Reviewer 2 Report
Dear authors,
The manuscript „Public Anxiety, Attitudes, and Practices towards COVID-19 Infection in Dammam, Saudi Arabia: A Cross-sectional Study” aims to research the Saudi community’s KAP towards COVID-19 infection in the city of Dammam. The manuscript is dealing with currently important topic, yet it needs some improvements.
In introduction section please correct that SARS-cov2 is previously known as COVID-19, actually it is the name of disease that this virus is causing. In sentence “As reported by [8],…” please include the name of first author before citing the reference number, and correct that throughout the manuscript. Please explain abbreviation when first used in manuscript (eg. KAP). It is not clear if the authors are investigating the knowledge of the participants in study. It is mentioned in aims, but is missing in the title of the manuscript as well as in the results. Please correct the aim of the study accordingly to address properly to the research that is conducted.
In methodology section please clarify what is represented in Figure 1. I suppose that those results are obtained by official authorities and should be clear that do not represent the results obtained in current manuscript.
Please clearly define sampling methods and justify the group recruited to take part in this study. Authors used self-developed scales designed for the purpose of the study. How was questionnaire designed? Was it pretested and in which population?
The sentence “The 95% confidence interval was considered, and the p-value < 0.05 was statistically significant.” should be rephrased to be easily understood.
Please explain in results and discussion the selection bias in recruiting the participants, since the majority is younger aged participant that could be result of easier access to the internet than older population….
Table 1 summarizes the demographic data of the participants, so results of infection and vaccination status as well as suffering of chronic disease do not belong to this category, please correct this. Furthermore, the Table 2 duplicates previous results presented in Table 1, in my opinion it should be better to include just Table 2, and to adjust it properly including correction according to comments given for Table 1.
In the end of the results and discussion section please include limitations of the study. The paper would gain from explicit and detailed discussion of limitations, not just to provide the list of shortcomings of your work. It is also important for you to explain how these limitations have impacted your research findings. Also, include couple of lines of future research.
The manuscript has some grammar and spelling errors that should be corrected.
The manuscript has some grammar and spelling errors that should be corrected.
Author Response
Reply to Review-2 Comments
Journal: Healthcare (ISSN 2227-9032)
Manuscript ID: healthcare-2474892
Type: Article
Title: Public Anxiety, Attitudes, and Practices towards COVID-19 Infection in Dammam, Saudi
Arabia: A Cross-sectional Study.
Authors: Mahmoud Mohamed Berekaa, Abdulaziz Abdulrahman AlMulla, Munthir Mohammed AlMoslem, Khalid Saif AlSahli, Mohammed Tawfiq AlJassim, Abdulmalik Salman AlSaif, Salman Ali AlQuwayi.
Point-by-point reply to reviewer-2 comments:
Reviewer-2 comments:
Dear authors,
The manuscript „Public Anxiety, Attitudes, and Practices towards COVID-19 Infection in Dammam, Saudi Arabia: A Cross-sectional Study” aims to research the Saudi community’s KAP towards COVID-19 infection in the city of Dammam. The manuscript deals with currently important topic, yet it needs some improvements.
- In the introduction section please correct that SARS-cov2 is previously known as COVID-19, actually it is the name of disease that this virus is causing.
Reply:
Thanks for your comment, sure I agree with your comment and modified as recommended.
- In sentence “As reported by [8],…” please include the name of first author before citing the reference number, and correct that throughout the manuscript.
Reply:
Thanks, corrected for reference 8 and checked throughout the manuscript as recommended.
- Please explain abbreviation when first used in manuscript (eg. KAP). It is not clear if the authors are investigating the knowledge of the participants in study. It is mentioned in aims but is missing in the title of the manuscript as well as in the results. Please correct the aim of the study accordingly to address properly to the research that is conducted.
Reply:
Thanks for your comment. In fact, assessment of Dammam Community for their knowledge about COVID-19 was not among the major aims of the study. The study focuses on anxiety, attitude and practices of the participants towards COVID-19 (non-infected and possibly infected).
On the other hand, the second part of the questionnaire included 17 COVID-19-related questions (personal and family history of COVID-19 infection, awareness degree of the main symptoms of COVID-19, attitude towards COVID-19 infection, and commitment to protective measures). This awareness section indirectly reflects their knowledge about the disease and the symptoms as well as the important protective measures. Moreover, emphasis was given to vaccination and most importantly on anxiety. Therefore, the knowledge was not intensively studied and discussed in the manuscript.
Finally, the major context of the study was revised throughout the manuscript to focus on the above-mentioned concepts.
- In the methodology section please clarify what is represented in Figure 1. I suppose that those results are obtained by official authorities and should be clear that do not represent the results obtained in the current manuscript.
Reply:
Thanks for your comment. Data source was already cited (Ref #33) with the figure legend. However, the same reference together with another reference cited within the text to provide more information about the COVID-19 infections in Eastern Province during the study period.
- Please clearly define sampling methods and justify the group recruited to take part in this study.
Reply:
Thanks for your comment. More information about the sample size and sampling methods were added.
- Authors used self-developed scales designed for the purpose of the study. How was the questionnaire designed? Was it pretested and in which population?
Reply:
Thanks for your comment. Additional data concerning the quality of the questionnaire was added. In fact, the validity of the questionnaire was determined by face validity and content validity, that were confirmed by three experts in public health. For reliability only Cronbach’s alpha was determined. Although alpha Cronbach value was 0.6 i.e. between 0.60 - 0.80 is considered moderate, but mostly acceptable as mentioned in Daud et al., 2018. (https://isdsnet.com/ijds-v7n3-15.pdf).
Daud, K.A.M., Khidzir, N.Z., Ismail, A.R. and Abdullah, F.A. (2018), “Validity and reliability of instrument to measure social media skills among small and medium entrepreneurs at Pengkalan Datu River”, International Journal of Development and Sustainability, Vol. 7 No. 3, pp. 1026-1037.
The sentence “The 95% confidence interval was considered, and the p-value < 0.05 was statistically significant.” should be rephrased to be easily understood.
Reply:
Thanks for your comment. Reformulated as recommended.
- Please explain in results and discussion the selection bias in recruiting the participants, since the majority is younger aged participant that could be result of easier access to the internet than older population….
Reply:
Really, thank you so much for your comment. I fact, this probable recruiting bias was mentioned as at the end of results and discussion section under the context of study limitations.
- Table 1 summarizes the demographic data of the participants, so results of infection and vaccination status as well as suffering of chronic disease do not belong to this category, please correct this.
Reply:
Thanks for your comment. In fact, this table covers the demographics of the participants and their important characteristics that strongly support the study.
Accordingly, the title of Table 1 changed from: “3.1. Demographic Characteristics of Participants” to “3.1. Demography and Characteristics of Participants”
- Furthermore, the Table 2 duplicates previous results presented in Table 1, in my opinion it should be better to include just Table 2, and to adjust it properly including correction according to comments given for Table 1.
Reply:
Thanks for your comment. It is important to mention that Table 2 is discussed in the context of the response of participants towards vaccination “The relationship between demographic characteristics and the vaccination”, and no duplication with table 1. Interestingly, the discussion was beyond the frequency and percentage; it also tests the chi-square and p-value to test the level of significance between variables, i.e. strengthen the study.
- In the end of the results and discussion section please include limitations of the study. The paper would gain from explicit and detailed discussion of limitations, not just to provide the list of shortcomings of your work. It is also important for you to explain how these limitations have impacted your research findings. Also, include couple of lines of future research.
Reply:
I really appreciate your comment.
I’d like to mention that I added two paragraphs at the end of the results and discussion section stating the major strengths and limitations in the study, highlighting the most of points you mentioned. The following paragraphs were added:
Current study provides an excellent opportunity to highlight the major importance of non-pharmaceutical measures to combat COVID-19 and similar diseases during early pandemics. The precautionary measures being implemented by Saudi government and healthcare agencies explain the remarkable decreased number of infected participants in comparison with many other countries. The study provides a great opportunity to assess the infected KAP towards COVID-19 coronavirus, in comparison with non-infected and other high-risk groups. Also, a major key feature that distinguishes the current study from others is the assessment of anxiety among COVID-19 infected and non-infected participants and in relation with demographic characteristics. Interestingly, the current study revealed that females showed higher anxiety scores than males. Also, public acceptance to vaccine was closely assessed and the study showed that old males’ groups were vaccinated more than females. In addition, the study unravels the relationship between being vaccinated and suffering from chronic diseases. Moreover, the study highlighted the importance of targeted health education intervention to increase public commitment to accept vaccines for future COVID-19 and other similar pandemics. There were optimistic attitudes and good practices among the Easter Province population in Saudi Arabia.
On the other hand, there are some limitations that should be considered during interpretation of the results. First, psychological fear or anxiety of infected participants to unravel their features during early pandemics. Second, the response and trust of participants towards vaccines might change during the context of pandemics due to massive governmental vaccinations and media interventions. Third, online questionnaires are mostly available for young people, therefore there is potential for participation bias. Finally, the results of KAP assessment of Dammam community in Easter Province cannot be generalized on national level.
- The manuscript has some grammar and spelling errors that should be corrected.
- Comments on the Quality of English Language
The manuscript has some grammar and spelling errors that should be corrected.
Reply:
Thanks for your comment. In fact, the manuscript has been proof-edited by SCRIBENDI (https://www.scribendi.com/academic.en.html) and the certificate of editing is available.

Reviewer 3 Report
Review of the manuscript "Public Anxiety, Attitudes, and Practices towards COVID-19 Infection in Dammam, Saudi Arabia: A Cross-sectional Study."
The manuscript aims to investigate the anxiety, attitudes, and practices of Eastern Province residents towards COVID-19 infection.
The manuscript presents interesting findings and aligns with the scope of MDPI Health care. However, it contains several imperfections that require major revisions before being considered for publication.
1. Title: I recommend aligning the title with the manuscript's structure. Therefore, I suggest changing it to "Public Anxiety, Attitudes, and Practices towards COVID-19 Infection in Eastern Province, Saudi Arabia: A Cross-sectional Study." Additionally, please mention the terminology of Dammam in the methodology section.
2. Abbreviations: Ensure that every abbreviation is defined at its first appearance in the abstract and introduction. Define COVID-19 and double-check that no abbreviations have been left undefined.
3. Abstract: In the "Results" section, avoid mentioning specific p-values and use "p" instead. Remove the reference to CHI (chi-square) in the results. Maintain consistency by following the pattern of reporting mean ± SD and p-values across all results.
4. Conclusion of the abstract: Provide more practical implications of the study findings in the abstract. Discuss how the results can contribute to improving public health measures or informing healthcare strategies related to COVID-19 in Eastern Province.
5. Keywords: Rephrase the keywords to avoid duplication with the title. Use MeSH (Medical Subject Headings) keywords, which are standardized terms used for indexing articles in the biomedical literature.
6. Introduction: There is some redundancy in the ideas presented in the introduction. Keep the aim of the study at the end of the introduction to enhance clarity. Please reorganize the introduction accordingly.
7. Methodology: To improve clarity, I recommend adding a flowchart that outlines the study protocol. This flowchart should provide a visual representation of the steps followed in the study, from participant recruitment to data analysis.
8. Sample Size Calculation: Include a section on sample size calculation in the methodology. Describe the rationale behind the chosen sample size and provide details of the statistical considerations used to determine it.
9. Results and Discussion: Split the results and discussion sections. Move the tables and figures exclusively to the results section. This will improve the structure and organization of the manuscript, allowing for a clear presentation of the findings before delving into their interpretation and discussion.
10. Strengths and Limitations: Include a dedicated section in the manuscript to discuss the strengths and limitations of the study. Highlight key strengths, such as a representative sample or the use of validated measurement tools. Additionally, discuss the limitations of the study, such as potential biases, generalizability issues, or any challenges encountered during data collection or analysis.
11. Declaration: Add a statement in the declaration section indicating whether AI was utilized at any stage of writing the manuscript. If applicable, provide a reference to the specific AI system used and refer to the paper you mentioned (https://pubmed.ncbi.nlm.nih.gov/37077800/) to acknowledge the use of AI technology.
12. Native English Review: The manuscript requires a thorough review by a native English speaker to enhance language quality. Emphasize that this is not a reflection of your English proficiency but rather a recognition of the need for improvement in the language used throughout the manuscript.
By addressing these points, the peer review report will be more fluent and coherent.
The manuscript requires a thorough review by a native English speaker to enhance language quality. Emphasize that this is not a reflection of your English proficiency but rather a recognition of the need for improvement in the language used throughout the manuscript.
Author Response
Reply to Review-3 Comments
Journal: Healthcare (ISSN 2227-9032)
Manuscript ID: healthcare-2474892
Type: Article
Title: Public Anxiety, Attitudes, and Practices towards COVID-19 Infection in Dammam, Saudi
Arabia: A Cross-sectional Study.
Authors: Mahmoud Mohamed Berekaa, Abdulaziz Abdulrahman AlMulla, Munthir Mohammed AlMoslem, Khalid Saif AlSahli, Mohammed Tawfiq AlJassim, Abdulmalik Salman AlSaif, Salman Ali AlQuwayi.
Point-by-point reply to reviewer-3 comments:
Reviewer-3 comments:
- Review of the manuscript "Public Anxiety, Attitudes, and Practices towards COVID-19 Infection in Dammam, Saudi Arabia: A Cross-sectional Study."
- The manuscript aims to investigate the anxiety, attitudes, and practices of Eastern Province residents towards COVID-19 infection.
- The manuscript presents interesting findings and aligns with the scope of MDPI Health care. However, it contains several imperfections that require major revisions before being considered for publication.
Reply:
We are so pleased to hear that the findings in the manuscript are well-aligned with the scope of the MDPI Healthcare journal. Indeed, your suggestions and comments to improve the manuscript are highly appreciated.
- Title: I recommend aligning the title with the manuscript's structure. Therefore, I suggest changing it to "Public Anxiety, Attitudes, and Practices towards COVID-19 Infection in Eastern Province, Saudi Arabia: A Cross-sectional Study." Additionally, please mention the terminology of Dammam in the methodology section.
Reply:
Thanks for your comment, the recommended changes have been made.
- Abbreviations: Ensure that every abbreviation is defined at its first appearance in the abstract and introduction. Define COVID-19 and double-check that no abbreviations have been left undefined.
Reply:
All abbreviations double checked and correctly defined in all manuscript especially during first appearance in abstract and introduction.
- Abstract: In the "Results" section, avoid mentioning specific p-values and use "p" instead. Remove the reference to CHI (chi-square) in the results. Maintain consistency by following the pattern of reporting mean ± SD and p-values across all results.
Reply:
I agree with you, the “p=value” was replaced by “p” in the abstract section. The reference to CHI (chi-square) was also deleted. Only mean ± SD, p-values and scores were consistently reported in the abstract section.
- Conclusion of the abstract: Provide more practical implications of the study findings in the abstract. Discuss how the results can contribute to improving public health measures or informing healthcare strategies related to COVID-19 in Eastern Province.
Reply:
The conclusion section in the abstract was modified to:
- Highlight the importance of the study especially reliable communication from health representatives and legislators to encourage knowledge of and trust in non-therapeutic interventions among the public.
- The needs for efficient intervention approaches to fill the gap during the implementation of non-therapeutic measures.
- Also, recommendation that awareness programs, during COVID-19 or any other similar pandemics, should be tailored to target Eastern Province inhabitants especially males.
- Keywords: Rephrase the keywords to avoid duplication with the title. Use MeSH (Medical Subject Headings) keywords, which are standardized terms used for indexing articles in biomedical literature.
Reply:
Your advice was highly appreciated; the keywords were rephrased to meet Medical Subject Headings (MeSH) standardized terms that is used for indexing articles in biomedical literature.
- 6. Introduction: There is some redundancy in the ideas presented in the introduction. Keep the aim of the study at the end of the introduction to enhance clarity. Please reorganize the introduction accordingly.
Reply:
Thanks for your comment. In fact, introduction was constructed and organized on a logical sequence started from, giving introduction about the origin and symptoms of the disease, with special emphasis on the needs for non-therapeutic interventions due to continuous genetic variation in the viral genome. The Second step was to discuss the importance of non-therapeutic interventions as a mean to combat the disease. The third and fourth sections highlighted the important research on global and national levels, respectively. Finally, the aim of the study was formulated at the end of the introduction section.
- Methodology: To improve clarity, I recommend adding a flowchart that outlines the study protocol. This flowchart should provide a visual representation of the steps followed in the study, from participant recruitment to data analysis.
Reply:
Thanks for your comment however, the study design a common analytical cross-sectional study that is mostly does not need characteristic flowchart model to explain as in the case of experimental epidemiological studies “clinical randomized trials”, where the CONSORT flow diagram is essential.
- Sample Size Calculation: Include a section on sample size calculation in the methodology. Describe the rationale behind the chosen sample size and provide details of the statistical considerations used to determine it.
Reply:
Thanks for your comment.
Concerning sample size calculations and methodology of data collection, the following paragraph was added:
The sample size was calculated using Cochran’s formula (Cochran, 2007). Data were collected from participants via an online questionnaire survey (using Google Forms), which was distributed through social media (WhatsApp, Telegram, and Twitter) using the convenience sampling technique. All adult (>18 years old) Dammam residents that can use media platforms and understand the content of the questionnaire were eligible to participate in the questionnaire survey for this study. Others unwilling to participate or unable to understand the content of the questionnaires are excluded.
Cochran WG: Sampling techniques. John Wiley & Sons; 2007.
- Results and Discussion: Split the results and discussion sections. Move the tables and figures exclusively to the results section. This will improve the structure and organization of the manuscript, allowing for a clear presentation of the findings before delving into their interpretation and discussion.
Reply:
Thanks for your comment. As authors, we assume that it is advantageous to merge results and discussion because each section in the results and discussion treated independently. Moreover, discussion of the results and findings in the same section would have greater impact than separation of them. Where, researchers can see the results, its presentation and enriched discussion in comparison with/to other researchers’ findings at the same section.
- Moreover, as mentioned in Instructions for Authors related to Healthcare journal, it is stated that there is possibility to combine discussion with the results as following:
- Discussion: Authors should discuss the results and how they can be interpreted in perspective of previous studies and of the working hypotheses. The findings and their implications should be discussed in the broadest context possible and limitations of the work highlighted. Future research directions may also be mentioned. This section may be combined with Results.
https://www.mdpi.com/journal/healthcare/instructions
- Strengths and Limitations: Include a dedicated section in the manuscript to discuss the strengths and limitations of the study. Highlight key strengths, such as a representative sample or the use of validated measurement tools. Additionally, discuss the limitations of the study, such as potential biases, generalizability issues, or any challenges encountered during data collection or analysis.
Reply:
Thanks for your comment. The following paragraphs were added:
Current study provides an excellent opportunity to highlight the major importance of non-pharmaceutical measures to combat COVID-19 and similar diseases during early pandemics. The precautionary measures being implemented by Saudi government and healthcare agencies explain the remarkable decreased number of infected participants in comparison with many other countries. The study provides a great opportunity to assess the infected KAP towards COVID-19 coronavirus, in comparison with non-infected and other high-risk groups. Also, a major key feature that distinguishes the current study from others is the assessment of anxiety among COVID-19 infected and non-infected participants and in relation with demographic characteristics. Interestingly, the current study revealed that females showed higher anxiety scores than males. Also, public acceptance to vaccine was closely assessed and the study showed that old males’ groups were vaccinated more than females. In addition, the study unravels the relationship between being vaccinated and suffering from chronic diseases. Moreover, the study highlighted the importance of targeted health education intervention to increase public commitment to accept vaccines for future COVID-19 and other similar pandemics. There were optimistic attitudes and good practices among the Easter Province population in Saudi Arabia.
On the other hand, there are some limitations that should be considered during interpretation of the results. First, psychological fear or anxiety of infected participants to unravel their features during early pandemics. Second, the response and trust of participants towards vaccines might change during the context of pandemics due to massive governmental vaccinations and media interventions. Third, online questionnaires are mostly available for young people, therefore there is potential for participation bias. Finally, the results of KAP assessment of Dammam community in Easter Province cannot be generalized on national level.
- Declaration: Add a statement in the declaration section indicating whether AI was utilized at any stage of writing the manuscript. If applicable, provide a reference to the specific AI system used and refer to the paper you mentioned (https://pubmed.ncbi.nlm.nih.gov/37077800/) to acknowledge the use of AI technology.
Reply:
Thanks for your comment. In fact, AI technology was not utilized at any stage of writing the manuscript (and not commonly applicable). Therefore, no need to add this statement.
- Native English Review: The manuscript requires a thorough review by a native English speaker to enhance language quality. Emphasize that this is not a reflection of your English proficiency but rather a recognition of the need for improvement in the language used throughout the manuscript.
Reply:
Thanks for your comment. In fact, the manuscript has been proof-edited by SCRIBENDI (https://www.scribendi.com/academic.en.html) and the certificate of editing is available.

Round 2
Reviewer 1 Report
Greetings,
In the attached document you will find the comments made.
Yours sincerely.

Author Response
Reply to Review-1 Comments
Journal: Healthcare (ISSN 2227-9032)
Manuscript ID: healthcare-2474892
Type: Article
Title: Public Anxiety, Attitudes, and Practices towards COVID-19 Infection in Dammam, Saudi
Arabia: A Cross-sectional Study.
Authors: Mahmoud Mohamed Berekaa, Abdulaziz Abdulrahman AlMulla, Munthir Mohammed
AlMoslem, Khalid Saif AlSahli, Mohammed Tawfiq AlJassim, Abdulmalik Salman AlSaif,
Salman Ali AlQuwayi.
Point-by-point reply to reviewer-1 comments:
Thank you for considering the comments made.
I only have two issues that I think should be looked at again.
- In the introduction, it was stated that it was necessary to add information on the status of COVID19 in the region described in the research at the time of study; in order to support or justify
the adoption of these measures. They state having added, "The total number of cases in the study
area was 6618 (out of 47235) during the period of the study, that represents 1.31% of all cases in
the kingdom during that period". This sentence is in results, not introduction and does not respond
to the request. Please add it in the introduction.
Reply:
Thanks for your comment.
- In fact, providing this information in the results section is more significant than being in introduction section because it explains the main reason for reduced number of infected candidates in the study “i.e. the major reason for the 5% infected candidates in the study”.
- Moreover, there is no significant impact on discussing these points in depth because it is highly affected by stringent intelligent lockdown in Saudi Arabia and accompanied measures and restrictions especially during the study period.
Thank you for addressing my concern and providing additional information on the quality of your
questionnaire. I understand that you have assessed content validity and face validity with the
support of public health experts, which is an important step to support the quality of the
instrument.
Reply:
Thanks for your comment. In fact, your comments helped to improve the quality of the study.
- As for the Cronbach's alpha coefficient obtained (0.61), I acknowledge that it is in the moderate
range and that some studies consider this level of internal consistency acceptable. However, given
that this coefficient is a commonly used measure to assess the reliability of a questionnaire, I would like to suggest that they conduct a more detailed analysis to better understand the sources
of variability and improve internal consistency.
In summary, I appreciate the effort you have made to address my comments and provide additional information on the quality of the questionnaire. However, I still consider that it would be beneficial to explore additional measures of reliability. This would help to strengthen the robustness of your study and provide a more comprehensive assessment of the quality of the
questionnaire used.
Reply:
Thanks for your comment. In fact, the use of Cronbach's alpha coefficient “that records 0.61” for testing the reliability and how closely related a set of items are as a group. In most cross-sectional studies, related to COVID-19 pandemic and other non-related research, this test is enough to express the degree of reliability as following:
Park, D.-I. Development and Validation of a Knowledge, Attitudes and Practices Questionnaire on COVID-19 (KAP COVID-19). Int. J. Environ. Res. Public Health 2021, 18, 7493. https://doi.org/10.3390/ ijerph18147493
Al-Hanawi MK, Angawi K, Alshareef N, Qattan AMN, Helmy HZ, Abudawood Y, Alqurashi M, Kattan WM, Kadasah NA, Chirwa GC, Alsharqi O. Knowledge, Attitude and Practice Toward COVID-19 Among the Public in the Kingdom of Saudi Arabia: A Cross-Sectional Study. Front Public Health. 2020 May 27;8:217. doi: 10.3389/fpubh.2020.00217. PMID: 32574300; PMCID: PMC7266869.
Salehi, A., Salmani, F., Norozi, E., Sadighara, P., & Zeinali, T. (2022). Knowledge, attitudes and practices of Iranian people about food safety and hygiene during COVID-19 pandemic. BMC Public Health, 22(1), 1148.
Ferdous MZ, Islam MS, Sikder MT, Mosaddek ASM, Zegarra-Valdivia JA, Gozal D (2020) Knowledge, attitude, and practice regarding COVID-19 outbreak in Bangladesh: An online-based cross-sectional study. PLoS ONE 15(10): e0239254. https://doi.org/10.1371/journal.pone.0239254
Luo, Y.-F.; Chen, L.-C.; Yang, S.-C.; Hong, S. Knowledge, Attitude, and Practice (KAP) toward COVID-19 Pandemic among the Public in Taiwan: A Cross-Sectional Study. Int. J. Environ. Res. Public Health 2022, 19, 2784. https://doi.org/10.3390/ijerph19052784
Kilinc, N. O., & Erci, B. (2022). Development and Psychometric Properties of Covid-19 Attitude Scale: For Health Workers. International Journal of Caring Sciences, 15(2), 1140.
And another non-related research:
Ekolu, S. O., & Quainoo, H. (2019). Reliability of assessments in engineering education using Cronbach’s alpha, KR and split-half methods. Global journal of engineering education, 21(1), 24-29.
aber, K.S. The Use of Cronbach’s Alpha When Developing and Reporting Research Instruments in Science Education. Res Sci Educ 48, 1273–1296 (2018). https://doi.org/10.1007/s11165-016-9602-2
I thank you for your attention and hope that these suggestions will be useful to further improve your research.
Reply:
Thanks so much, your comments and suggestion have great impact on improvement of our study.

Reviewer 2 Report
good job!
Author Response
Reply to Review-2 Comments
Journal: Healthcare (ISSN 2227-9032)
Manuscript ID: healthcare-2474892
Type: Article
Title: Public Anxiety, Attitudes, and Practices towards COVID-19 Infection in Dammam, Saudi
Arabia: A Cross-sectional Study.
Authors: Mahmoud Mohamed Berekaa, Abdulaziz Abdulrahman AlMulla, Munthir Mohammed AlMoslem, Khalid Saif AlSahli, Mohammed Tawfiq AlJassim, Abdulmalik Salman AlSaif, Salman Ali AlQuwayi.
Point-by-point reply to reviewer-2 comments:
Reviewer-2 comments:
|
Yes |
Can be improved |
Must be improved |
Not applicable |
|
|
Does the introduction provide sufficient background and include all relevant references? |
(x) |
( ) |
( ) |
( ) |
|
Are all the cited references relevant to the research? |
(x) |
( ) |
( ) |
( ) |
|
Is the research design appropriate? |
(x) |
( ) |
( ) |
( ) |
|
Are the methods adequately described? |
(x) |
( ) |
( ) |
( ) |
|
Are the results clearly presented? |
(x) |
( ) |
( ) |
( ) |
|
Are the conclusions supported by the results? |
(x) |
( ) |
( ) |
( ) |
Comments and Suggestions for Authors
good job!
Reply:
Thanks so much for your comments and suggestions. Your feedback helped us to greatly improve the manuscript.

Reviewer 3 Report
Firstly, I would like to extend my congratulations to the authors for this revised version of the manuscript, which is a significant improvement over the previous one. The article is now ready for publication, but I do have a minor comment regarding the title. I suggest it be revised to: "Public Anxiety, Attitudes, and Practices towards COVID-19 Infection in an Eastern Province of Saudi Arabia: A Cross-sectional Study".
Average
Author Response
Reply to Review-3 Comments
Journal: Healthcare (ISSN 2227-9032)
Manuscript ID: healthcare-2474892
Type: Article
Title: Public Anxiety, Attitudes, and Practices towards COVID-19 Infection in Dammam, Saudi
Arabia: A Cross-sectional Study.
Authors: Mahmoud Mohamed Berekaa, Abdulaziz Abdulrahman AlMulla, Munthir Mohammed AlMoslem, Khalid Saif AlSahli, Mohammed Tawfiq AlJassim, Abdulmalik Salman AlSaif, Salman Ali AlQuwayi.
Point-by-point reply to reviewer-3 comments:
Reviewer-3 comments:
Comments and Suggestions for Authors
Firstly, I would like to extend my congratulations to the authors for this revised version of the manuscript, which is a significant improvement over the previous one. The article is now ready for publication, but I do have a minor comment regarding the title. I suggest it be revised to: "Public Anxiety, Attitudes, and Practices towards COVID-19 Infection in an Eastern Province of Saudi Arabia: A Cross-sectional Study".
Comments on the Quality of English Language
Average
Reply:
First, thank you so much for your efforts, your suggestions and comments helped us to greatly improve the manuscript are highly appreciated.
- Concerning the manuscript title: your suggestion highly appreciated, and it will be changed to "Public Anxiety, Attitudes, and Practices towards COVID-19 Infection in an Eastern Province of Saudi Arabia: A Cross-sectional Study" as per your recommendation.
- The manuscript subjected to further English language revision as recommended.
